# Mental Health during COVID-19 Pandemic among Caregivers of Young Children in Kenya’s Urban Informal Settlements. A Cross-Sectional Telephone Survey

**DOI:** 10.3390/ijerph181910092

**Published:** 2021-09-25

**Authors:** Vibian Angwenyi, Margaret Kabue, Esther Chongwo, Adam Mabrouk, Ezra Kipngetich Too, Rachel Odhiambo, Carophine Nasambu, Joyce Marangu, Derrick Ssewanyana, Eunice Njoroge, Eunice Ombech, Mercy Moraa Mokaya, Emmanuel Kepha Obulemire, Anil Khamis, Amina Abubakar

**Affiliations:** 1Institute for Human Development, Aga Khan University, Nairobi P.O. Box 30270-00100, Kenya; margaret.kabue@aku.edu (M.K.); esther.chongwo@aku.edu (E.C.); adam.mabrouk@aku.edu (A.M.); ezra.too@aku.edu (E.K.T.); rachel.odhiambo@aku.edu (R.O.); joyce.marangu@aku.edu (J.M.); ssewanyana@lunenfeld.ca (D.S.); eunice.njoroge@aku.edu (E.N.); eunice.ombech@aku.edu (E.O.); mercy.mokaya@aku.edu (M.M.M.); emmanuel.obulemire@aku.edu (E.K.O.); anil.khamis@aku.edu (A.K.); amina.abubakar@aku.edu (A.A.); 2Neurosciences Group, KEMRI/Wellcome Trust Research Programme, Centre for Geographic Medicine Research (Coast), Kenya Medical Research Institute, Kilifi P.O. Box 230-80108, Kenya; CNasambu@kemri-wellcome.org; 3Alliance for Human Development, Lunenfeld-Tanenbaum Research Institute, Toronto, ON M5T 3L9, Canada; 4Institute of Education, University College London, 20 Bedford Way, London WC1H 0AL, UK

**Keywords:** COVID-19, COVID-19 anxiety scale, general anxiety disorder-7 scale, Kenya, mental health, patient health questionnaire-9, urban informal settlements, telephone cross-sectional survey

## Abstract

The emergence of COVID-19 has profoundly affected mental health, especially among highly vulnerable populations. This study describes mental health issues among caregivers of young children and pregnant women in three urban informal settlements in Kenya during the first pandemic year, and factors associated with poor mental health. A cross-sectional telephone survey was administered to 845 participants. Survey instruments included the Patient Health Questionnaire-9, General Anxiety Disorder-7 scale, COVID-19 Anxiety Scale, and questions on the perceived COVID-19 effects on caregiver wellbeing and livelihood. Data were analyzed using descriptive statistics, and univariate and multivariate analysis. Caregivers perceived COVID-19 as a threatening condition (94.54%), affecting employment and income activities (>80%). Caregivers experienced discrimination (15.27%) and violence (12.6%) during the pandemic. Levels of depression (34%), general anxiety (20%), and COVID-19 related anxiety (14%) were highly prevalent. There were significant associations between mental health outcomes and economic and socio-demographic factors, violence and discrimination experiences, residency, and perceptions of COVID-19 as a threatening condition. Caregivers high burden of mental health problems highlights the urgent need to provide accessible mental health support. Innovative and multi-sectoral approaches will be required to maximize reach to underserved communities in informal settlements and tackle the root causes of mental health problems in this population.

## 1. Introduction

The emergence of COVID-19, which was first reported in Wuhan China in December 2019, has had far-reaching impact globally [1]. By March 2020, many countries across the globe had reported their first COVID-19 cases or were in the peak of the first wave of the epidemic [2]. To address the escalating COVID-19 cases and contain the pandemic, the World Health Organization (WHO) provided guidance on containment measures for countries to adopt and declared COVID-19 a global public health emergency [3]. The COVID-19 mitigation measures, such as lockdowns, travel restrictions, quarantine, and isolations, have contributed to job losses, social and daily living disruptions, and elevated levels of stress and anxiety, which have a profound effect on mental health [4,5]. The overflow of COVID-19 related (mis)information in mainstream and social media platforms has been reported to trigger fear and panic, and negatively affecting the observance of COVID-19 measures [6]. More importantly, there is the likelihood that some people may project excessive fear of having the coronavirus disease (also termed as pathological health anxiety), altering their behaviors by relentlessly concerning and being preoccupied with safety seeking behavior [7]. The uncertainties and unpredictable nature of the COVID-19 pandemic could result to individuals projecting various forms of trauma and stress-disorders, such as depression, anxiety disorders, psychosocial dysfunction, dissociation disorders, substance abuse, and insomnia, among others [5,8,9]. It is projected that these effects will be most severe in low resourced settings seeking to address these emerging mental health needs, in the context of weak health systems with limited access to mental health services [4]. Moreover, it is important to underscore the pandemic’s effect on frontline service providers (e.g., increased workload, burnout, and stress, the risk of COVID-19 infections, and stigma associated with being in contact with COVID-19 patients), and their resilience to offer continued care in such difficult circumstances [7,10].

Kenya reported its first COVID-19 case on the 12 March 2020 and by 28 June 2021 confirmed cases had risen to over 182,000 [2], with some informal settlements within Kenya’s capital city Nairobi (such as Kawangware and Kibera) identified as epidemic ‘hot spots’ for community transmission [8]. The Kenyan government enforced several control measures, such as immediate closure of learning institutions, dusk-to-dawn curfews, stay at home measures, mandatory wearing of face masks in public with a hefty penalty for non-compliance [11,12]. For families in informal settlements, observing these measures may be more complex and their ability to care for their young children during the pandemic compromised. Such families exhibit unstable sources of income and the high dependence on casual jobs with daily wages, and threatened livelihoods and food security. Informal settlement dwellings are poorly constructed and crowded with poor sanitation infrastructure making it difficult to enforce COVID-19 preventive measures, such as physical distancing and regular handwashing practices [11]. These conditions are likely to compromise mental health functioning for those living in urban informal settlements. Our interest for focusing on caregivers of young children were two-fold. First, urban informal settlements comprise an assortment of individuals (e.g., low income earners, asylum seekers, refugees), who are at high risk of mental health problems due to multiple stressors and this may be further aggravated by the COVID-19 pandemic. Second, due to the closure of learning institutions for prolonged period of time, most caregivers had to juggle between work, childcare, and deal with COVID-19 related stressors, which may have elevated their risk for developing mental health problems. Furthermore, the impact of mental illness and distress in caregivers may have serious implications for their health, as well as the growth and development of their young children, hence it raises the need to identify and address mental health issues for this sub-population and their potential influence on future outcomes.

Prior to the pandemic, research has established that pregnant women experience anxiety and depressive symptoms, estimated to be as high as 15–40%, across all trimesters. [13]. In a recent review by Shorey et al. [14], the pooled prevalence of antenatal anxiety and depressive symptoms during COVID-19 period were reported to be high (40% and 27%, respectively), although most of these studies were based in high income countries and Asia. The pandemic and implementation of COVID-19 containment measures will greatly affect perinatal health [14,15,16]. For instance, the fear of COVID-19 vertical transmission to unborn babies may affect pregnant women’s access to antenatal care, lockdowns and movement restrictions may cut-off access to social support, and physical distancing causing social isolation, which may contribute to elevated stress and anxiety [15,16]. Mental health problems experienced during pregnancy have detrimental implications for self-care and health seeking practices [17], and have an effect on childbirth, and child and parenting outcomes [18]. For instance, the increased risk of pre-eclampsia, preterm births, childhood cognitive, and behavioral problems (e.g., infant negative emotional reactivity) and impaired attachment [18]. Hence, this raises the need to have a better understanding of issues affecting pregnant women during the pandemic period and especially those from low-resourced settings such as Africa’s urban informal settlements, whose situation may be worsened due to the prevailing social, economic, and health infrastructure disparities.

There is, therefore, a need to describe mental health status for those most vulnerable, especially in the informal settlements during the COVID-19 pandemic, with a view to identify the magnitude of the problem and identify strategic points of intervening. The study aim was to describe mental health problems experienced by caregivers of young children below five years and pregnant women in three urban informal settlements in Nairobi and Mombasa, Kenya. Furthermore, the study explored factors associated with poor mental health during the COVID-19 pandemic and provides recommendations to support caregivers.

## 2. Materials and Methods

### 2.1. Study Design and Data Sources

This paper is based on a sequential mixed methods study following the approach described in Creswell et al. [19]. The main study aim was to examine the psychosocial functioning, economic well-being, and mental health during the COVID-19 pandemic among caregivers of young children in Kenya’s urban informal settlements. The first phase was a telephone survey among caregivers of children below five years (with and those without disabilities), and pregnant women to characterize the mental health burden and the extent of COVID-19 effect on their wellbeing and families. The second phase was a qualitative inquiry to explore further the effects of COVID-19 on health, child care, and mechanisms employed to cope and respond to the pandemic. The focus of this paper is on the telephone survey among caregivers conducted between 23 September and 22 October 2020, a time when some of the COVID-19 restrictions had eased (e.g., in-country travel, curfew hours shortened, and partial opening of learning institutions).

### 2.2. Study Setting

This study was conducted in three urban informal settlements: Bangladesh in Mombasa County; Mathare and Dagoretti’s informal settlements (e.g., Kawangware) in the Nairobi Metropolitan Area. According to the UN-Habitat, informal settlements are residential areas where housing standards do not comply with building regulations and often situated in environmentally hazardous areas, neighbourhoods lack formal services, and there is no security of tenure in the land occupied [20]. Bangladesh is one of the oldest and largest informal settlements in Mombasa, Jomvu sub-county [21]. Bangladesh residents migrated primarily from rural areas and other parts of the coastal region, seeking economic opportunities within Mombasa region [21]. Bangladesh borders several industries where residents seek casual employment and also engage in small income generating activities. The settlement has a population of over 20,000 people with nearly 2800 households [21]. Housing situation in Bangladesh is typical to other informal settlements in Kenya.

Mathare is one of the oldest informal settlements in Nairobi, within Ruaraka sub-county, with an estimated population of nearly 206,564 in 2019 [22]. It is a relatively disadvantaged area with more than half the population living in a house without a concrete floor, with no permanent walls and having to share toilet facilities. The open and unsafe disposal of faecal waste in this setting has been linked to frequent outbreaks of infectious diseases [23].

Kawangware is a low-income residential area in Dagoretti sub-county, about 12 km west of Nairobi Central Business District. Dagoretti sub-county population was 434,208 in 2019 [22], with a sizeable refugee presence. Most refugees and asylum seekers in the settlement are from the Great Lakes region, the Horn of Africa, and the Democratic Republic of Congo [24]. The living conditions in Dagoretti’s informal settlements are considered relatively better compared to other informal settlements in Nairobi. Despite this, there are many crowded shacks, with limited access to piped water, deficient sewage system, and high crime rate due to unemployment [24].

### 2.3. Participants and Procedures

Non-probability and purposive sampling approaches were used in this study. Through consultations with the sub-county community health focal persons, we identified 56 community health units with households of low socio-economic status in the three informal settlements. Community health volunteers (CHVs) from these community units were involved to support with identification of households meeting eligibility criteria, and the sample summary and recruitment status by site is illustrated in Figure 1. The survey inclusion criteria included having a child below five years (with or without disabilities), being currently pregnant, above 18 years, and a resident of these informal settlements. A decline to provide informed consent was the only exclusion criteria. Analysis of data from caregivers of young children below five years with disabilities (n = 165) is reported in a separate manuscript, since this group experiences multiple vulnerabilities that would require in-depth exploration and discussion.

CHVs were provided with simplified key messages and data capture forms to collect basic demographic and contact details of eligible participants. In compliance with COVID-19 safety measures, the research team relied on CHVs to conduct home visits and provided them with sufficient personal protective equipment (PPE). Thereafter, a team of 11 trained research assistants and field enumerators contacted identified eligible participants on phone for their consent to be involved and survey administration.

### 2.4. Measures and Data Collection Procedures

To assess mental health, we utilized the 9-item Patient Health Questionnaire (PHQ-9), the 7-item General Anxiety Disorder (GAD-7), and the 5-item COVID-19 Anxiety Scale (CAS)—see Appendix A. The Swahili version of the PHQ-9 has been adapted and validated among Kenyan adults as a 9-item scale for screening for depression [25]. The PHQ-9 scale possessed good internal consistency (Macdonald’s omega > 0.80), high test-retest reliability (ICC-0.64), and found to be a unidimensional scale with good discriminant validity [25].

Similarly, the Swahili version of the GAD-7 scale has been adapted and validated among the adult Kenyan population for screening for general anxiety disorder [26]. The internal consistency of the Swahili GAD-7 was found to be high (Cronbach’s alpha = 0.82), had an acceptable test-retest reliability (ICC = 0.7), and based on confirmatory factor analysis, it was found to be a unidimensional scale with good discriminant validity [26]. In this study, the internal consistency score was excellent (Cronbach’s alpha 0.84 (0.83–0.86) and Omega 0.85 (0.82–0.87)).

The COVID-19 anxiety scale is a newly developed measure to screen for clinical anxiety and fear associated with the coronavirus disease [16]. The five-item scale measures dizziness, sleep disturbance, tonic immobility, appetite loss, and abdominal distress on a five-point scale from 0 (not at all) to 4 (nearly every day) based on symptoms over the past two weeks [17]. Although CAS is still a novel measure, it has been adapted and validated for use among adults in various languages and contexts around the world, e.g., in Brazil, Turkey, and India, and found to have acceptable internal consistency, high test-rest reliability and to fit a one-factor solution, i.e., being unidimensional [27,28,29]. In the present study, we followed an adaptation process (e.g., forward and back translation, cognitive interviewing) to ensure that the Swahili version of CAS was culturally appropriate and relevant for the adult sample in our study context. Psychometric analysis performed on the Swahili version of the CAS had high internal consistency (Cronbach’s alpha = 0.87), and the confirmatory analysis of a one-factor solution of the CAS showed an excellent fit to the hypothesized structure (RMSEA =  0.00, CFI = 1.00, TLI = 1.00), and all the items had a factor loading of above 0.40. Furthermore, the scores indicated convergent validity since they were correlated with PHQ scores (r = 0.62) and GAD scores (r = 0.57).

Other variables of interest collected in the survey were demographic characteristics (gender, age, marital status, education, occupation, number of children (including if expecting a child/currently pregnant)). Additionally, we included questions with Likert-scale and ‘Yes/No’ response options to explore: (1) the perceived threat of COVID-19 on health and household wellbeing; (2) experiences of discrimination during COVID-19 period; (3) experiences of violence during COVID-19 period; and (4) perceived effect of COVID-19 on economic and livelihood of households. Most of these aspects were selected because they are plausible factors that interplay in the mechanisms through which COVID-19 is likely to elevate mental illness in the community and households. All these items were assessed using the COVID-19 pandemic as the reference period.

Participant’s socioeconomic status was assessed using a 9-asset index, which included questions on items available in people’s homes, e.g., bicycle, television, motorbike, refrigerator, mobile phone, radio, motor vehicle, and internet connectivity. The asset index is a proxy measure of household standards of living by summing up commonly found household assets and categorizing individuals in different wealth quintiles, from poorest to richest based on overall score [30]. This is a commonly used measure in demographic health surveys [31], and in Kenya, it has also been used by the government social protection programmes to identify vulnerable and poor households. In this study, a score of one was assigned to ownership of each asset and zero score for lack of ownership, based on the standard approach [30]. However, motor vehicle ownership was assigned three points due to its high market value, which was a slight modification for our study. A total score was then generated. A higher asset index score indicates a higher socio-economic status and vice versa.

The survey instrument was pre-tested via telephone among 14 respondents from two of the three urban informal settlements studied (Mathare and Dagoretti). These informants were selected purposively and identified by community health volunteers to represent caregivers of young children below five years (with or without disabilities), and pregnant women. The pre-test was mainly to check respondent’s comprehension of questions, translations of scale items into Kiswahili, instrument coherence, duration to complete the survey, and the practicalities of administering the survey on telephone. The survey was anonymous and researcher-administered (by five males and six females), who read out all questions and response category options (e.g., Likert-scales), which were captured electronically using the Open Data Kit (ODK) platform. Median time to complete the survey was 43 min (20–58 inter-quartile range). Completed survey forms were verified prior to uploading onto a secure web-based database. Anonymity was observed during data collection by ensuring that research assistants only had access to anonymized records and the study data manager removed all identifiers in the dataset used for analysis. Phone recordings and electronic survey forms were later archived and stored within the authors institution and were only accessible to the research team.

### 2.5. Statistical Analysis

Data were analyzed using STATA (Version 15; StataCorp). Percentages and frequencies were used to describe participant’s socio-demographic characteristics. Pearson’s chi-squared test was used to assess group differences in the categorical variables, while an independent *t*-test was used to analyze group differences for continuous variables. Univariate logistic regression was used to examine relationships between independent variables and mental health status (depression, COVID-19 related anxiety, and general anxiety disorders). A cut-off score of <10 was used for PHQ-9, <9 for COVID-19 anxiety scale, and <10 for generalized anxiety disorder scale. These cut-off scores are the recommended (standardized) cut-off points. Variables with a *p* < 0.20 from the univariate logistic regression were fitted into the multivariable logistic regression model. Multicollinearity was checked by conducting correlational analysis. Odds ratios (ORs), 95% confidence intervals (C.I), and associated P-values were reported. A *p*-value of <0.05 was considered statistically significant. Tables were used to present these findings.

## 3. Results

### 3.1. Socio-Demographic Characteristics and Mental Health Status

A total of 1262 participants were approached for the survey, with 80.3% (n = 1010) response rate, while those who declined consent to participate were 11 out of the 252 excluded, as illustrated in Figure 1. We draw on data from 845 participants. Participant’s mean age was 29.00 (SD 7.15) years. The majority of caregivers were female 801 (94.79%), married or cohabiting 597 (71.17%), had a secondary school education 379 (44.85%), were unemployed 415 (49.11%), and caring for less than four children of all ages 623 (79.97%)—see Table 1. There were no significant differences across the three informal settlements by participant’s age, gender, occupation, marital status, number of children, and pregnancy status. However, there were significant differences across the three sites by education level (*p* < 0.001) and total asset index score (*p* < 0.001). Compared to participants from Bangladesh and Mathare, the caregivers from Dagoretti’s informal settlements had higher levels of education and household assets—see Supplementary File S4.

In this study, more than one-third (34.08%) of caregivers reported having depressive symptoms, 13.96% experienced COVID-19 related anxiety, and 19.67% reported experiencing generalized anxiety. Additionally, severe forms of depressive symptoms were reported in 14.20% (n = 120) of the caregivers, while 7.93% (n = 67) experienced severe generalized anxiety—see Appendix A.

There were no significant differences in age, gender, education level, occupation, and pregnancy status among caregivers with depressive symptoms, COVID-19 related anxiety, and generalized anxiety. Notably, amongst those with depressive symptoms, COVID-19 related anxiety, and general anxiety, the majority were females (i.e., 93.75%, 97.46% and 93.41%, respectively—see Table 1). We found significant differences for all three mental health outcomes by household socio-economic status (PHQ *p* < 0.001; CAS *p* = 0.037; GAD *p* = 0.047) and area of residency or study site (PHQ *p* < 0.001; CAS *p* < 0.001; GAD *p* = 0.003). The proportion of respondents with depressive symptoms was higher for those in a marital union (67%) compared to those single/separated (*p* = 0.012)—see Table 1.

### 3.2. COVID-19-Related Consequences on Psychosocial Aspects

Caregivers largely perceived COVID-19 as a serious threat to their health and households (94.54%), reported to have lost their jobs (82.31%), and reported a loss in income generating activities (81.62%) due to the pandemic. Similarly, some caregivers reported that they experienced discrimination (15.27%), and violence (12.66%) during the pandemic period. Additionally, most of the caregivers reported that their ability to cater for basic utilities such as rent (83.87%), paying loans (77.23%), and their ability to cater for their childcare needs (77.61%) were severely affected due to the COVID-19 mitigation measures implemented by the government. Caregivers further reported that during the COVID-19 pandemic, movement outside households (n = 478; 56.77%) and family interactions (n = 552; 65.56%) were affected to a large extent (see Table 2).

Generally, a significantly higher proportion of mental health issues (depressive symptoms, COVID-19 related anxiety, and general anxiety) were reported by caregivers who perceived that COVID-19 mitigation measures, to a greater extent, affected their jobs and other income activities, as well as interactions with family and other people outside their households (*p* < 0.05). For instance, there was a significantly higher proportion of caregivers with depressive symptoms who reported that COVID-19 affected their jobs as compared to caregivers without depressive symptoms (92.8% VS. 76.6%, *p* < 0.001)—see Table 2 for detailed account of these characteristics.

### 3.3. Correlates of Mental Health Outcomes

#### 3.3.1. Univariate Analysis

In the univariate analysis, being separated, divorced, or widowed (UnadjOR, 1.78, 95% CI (1.2–2.63)), lower household asset index score (UnadjOR, 0.74, 95% CI (0.65–0.84)); area of residency—that is, living in Bangladesh (UnadjOR, 2.07, 95% CI (1.49–2.86)), experiencing violence (UnadjOR, 2.86, 95% CI (1.89–4.31)), and discrimination (UnadjOR, 2.92, 95% CI (2.00–4.29)), loss of job (UnadjOR, 3.96, 95% CI (2.37–6.60)) and income generating activities (UnadjOR, 2.30, 95% CI (1.46–3.64)) inability to afford basic childcare needs (UnadjOR, 1.47, 95% CI, (1.10–1.97)), restricted movement outside households (UnadjOR, 1.47, 95% CI (1.10–1.97)) and interaction with family members affected during the pandemic (UnadjOR, 1.66, 95% CI (1.21–2.26)), were factors significantly associated with experiencing depressive symptoms—see Appendix A.

For COVID-19 related anxiety, univariate logistic regression showed the following covariates were significant risk factors: higher education level (UnadjOR, 2.30, 95% CI (1.01–5.20)); lower household asset index score (UnadjOR, 0.82, 95% CI (0.69–0.98)); area of residency, i.e., living in Bangladesh (UnadjOR, 2.88, 95% CI (1.86–4.46)); experiencing violence (UnadjOR, 2.11, 95% CI (1.29–3.48)) and discrimination (UnadjOR, 2.55, 95% CI (1.62–4.03)) during the pandemic; COVID-19 perceived as a serious threat (UnadjOR, 7.67, 95% CI (1.05–56.15)); loss of job (UnadjOR, 2.40, 95% CI (1.22–4.73)) and income generating activities (UnadjOR, 2.13, 95% CI (1.11–4.11)); and restricted movement outside households (UnadjOR, 1.70, 95% CI (1.12–2.57)) and interaction with family members affected (UnadjOR, 1.56, 95% CI, (1.01–2.42)) during the pandemic—see Appendix A.

For generalized anxiety, univariate logistic regression analysis showed that having more than 3 children per household (UnadjOR, 1.54, 95% CI (1.02–2.32)), lower household asset index score (UnadjLR, 0.86, 95% CI (0.74–1.00)), area of residency, i.e., living in Bangladesh (UnadjOR, 1.67, 95% CI (1.14–2.43)), experiencing violence (UnadjOR, 2.12, 95% CI (1.35–3.32)) and discrimination (UnadjOR, 3.24, 95% CI (2.16–4.86)), COVID-19 perceived as a serious threat (UnadjOR, 3.69, 95% CI (1.13–12.03)), loss of job (UnadjOR, 1.78, 95% CI (1.05–3.03)) and income generating activities (UnadjOR, 1.86, 95% CI (1.07–3.21)), inability to meet basic childcare needs (UnadjOR, 1.74, 95% CI (1.14–2.67)), restricted movement outside households (UnadjOR, 1.59, 95% CI (1.12–2.27)) and interaction with family members affected (UnadjOR, 1.73, 95% CI (1.18–2.54)) during the pandemic were factors associated with generalized anxiety—see Appendix A.

#### 3.3.2. Multivariate Analysis

Results from the multivariate logistic regression analysis as illustrated in Table 3, showed that caregivers who experienced violence were twice as likely to experience depressive symptoms (AdjOR, 1.89, 95% CI, 1.12–3.18; *p* < 0.016) compared to caregivers who did not experience violence. Similarly, caregivers who experienced discrimination were thrice as likely to experience depressive symptoms compared to caregivers who did not experience discrimination (AdjOR, 3.12; 95% CI, 1.91–5.10; *p* < 0.001). Furthermore, the odds of experiencing depressive symptoms were three times greater among caregivers who had lost their jobs during the pandemic compared to those whose jobs were not affected (AdjOR, 2.88; 95% CI, 1.51–5.48; *p* = 0.001). Caregivers living in Bangladesh were twice as likely to experience depressive symptoms compared to caregivers in Dagoretti’s informal settlements (AdjOR, 1.93; 95% CI, 1.26–2.98; *p* = 0.003). We found an association between socio-economic status and experiences of depression; that is, a unit increase in caregiver’s home asset was significantly associated with a decrease in the odds of experiencing depressive symptoms (AdjOR, 0.83; 95% CI, 0.69–0.99; *p* = 0.041).

For COVID-19 related anxiety, lower levels of education, area of residency, experiencing discrimination and violence, and inability to pay for basic utilities were associated with COVID-19 related anxiety. Compared to those who did not experience discrimination, those who experienced discrimination were twice as likely to report COVID-19 related anxiety (AdjOR, 2.35; 95% CI, 1.35–4.09; *p* = 0.002)—see Table 3. Caregivers with education attainment of primary level and below were thrice as likely to report COVID-related anxiety when compared to those with tertiary level education (AdjOR, 2.79; 95% CI, 1.06–7.36; *p* = 0.038). Similar to the PHQ-9 scale, caregivers living in Bangladesh were twice as likely to report COVID-19 related anxiety when compared to caregivers in Dagoretti’s informal settlements (AdjOR, 2.12; 95% CI, 1.23–3.65; *p* = 0.007). The odds of experiencing COVID-19 related anxiety were two times greater among caregivers who experienced violence compared to those who did not experience violence (AdjOR, 1.83; 95% CI, 1.01–3.32; *p* = 0.048). The odds of experiencing COVID-19 related anxiety were two times greater among caregivers who experienced violence compared to those who did not experience violence (AdjOR, 2.35; 95% CI, 1.35–4.09; *p* = 0.002).

For generalized anxiety, the multivariate logistic regression analysis showed caregivers in formal employment were thrice as likely to experience generalized anxiety compared to unemployed caregivers (AdjOR, 2.84; 95% CI, 1.21–6.63; *p* = 0.016). The odds of experiencing generalized anxiety was two times greater among caregivers who experienced violence compared to those who did not experience violence (AdjOR, 1.80; 95% CI, 1.03–3.13; *p* = 0.037), and four times greater among caregivers who experienced discrimination compared to those who did not experience discrimination (AdjOR, 3.62; 95% CI, 2.19–5.98; *p* < 0.001). The ability to pay for basic utilities and repay loans were negatively associated with generalized anxiety (AdjOR, 0.34; 95% CI, 0.14–0.82; *p* = 0.016) and (AdjOR, 0.48; 95% CI, 0.25–0.91; *p* = 0.024), respectively. This is further illustrated in Table 3.

## 4. Discussion

### 4.1. Key Findings

During the first year of the COVID-19 pandemic, caregivers of young children living in Kenya’s urban informal settlements who participated in the survey experienced a high burden of mental health problems [depressive symptoms (34%), generalized anxiety (14%), and COVID-19 related anxiety (20%)]. Several factors were associated with poor mental health among caregivers during the pandemic. Multivariate regression analysis shows that experiences of violence and discrimination are factors associated with experiencing depressive symptoms, generalized anxiety, and COVID-19 related anxiety. Loss of employment during the pandemic increased the likelihood of experiencing depressive symptoms by two-fold. Our analysis further shows informal settlement differences in mental health status; that is, caregivers in Bangladesh had higher depressive symptoms and COVID-19 related anxiety scores compared to caregivers in Dagoretti’s informal settlements. This could possibly be explained by contextual differences; that is some settings could be experiencing more hardship than others partly due to varying socio-economic levels (e.g., Dagoretti has relatively better standards of living compared to other informal settlements), varying infrastructure levels, and resource access. More research is needed to investigate mechanisms that explain variation in the mental health status in these settings. Findings show that a relative increase in household asset index score (proxy of wealth) and higher educational attainment lowered the odds of experiencing depressive symptoms and COVID-19 related anxiety, respectively. Unexpectedly, caregivers in formal employment are more likely to report anxiety when compared to those unemployed or in the informal sector. A possible explanation for this observed finding within our study population could be that caregivers in formal employment may have been more apprehensive about safeguarding their jobs, since during the pandemic there were drastic job cuts in the formal sector witnessed in the country [32]. Though of equally great concern was the effect of the COVID-19 pandemic on Kenya’s informal sector economy, and this was also observed in the same report [32].

Several additional factors significantly associated with poor mental health were identified in the univariate regression analysis. Findings reveal that the majority of caregivers who perceived COVID-19 as a very serious threat, and experienced restricted family interaction and movement outside households during the pandemic had higher depressive symptoms, general anxiety, and COVID-19 related anxiety. Caregivers reporting COVID-19 affected to a great extent their ability to meet basic childcare needs experienced depressive symptoms and anxiety. Caregivers who were separated, divorced, or widowed and were caring for more children (>3) were more likely to experience depression and anxiety. Our analysis did not reveal any significant associations between mental health status with age, gender, and being pregnant. However, we found significant site differences with regard to caregivers’ level of education and household asset index (proxy of wealth).

### 4.2. Implication of Findings and Proposed Recommendations

Caregivers in informal settlements have been severely affected by loss of employment and income as a result of the COVID-19 pandemic, aggravated by the impoverished living conditions and poor general infrastructure. This situation experienced by caregivers has critical implications for themselves and the wellbeing of their children. That is, an inability to optimally cater to family needs, especially for young children, and following COVID-19 mitigation measures within these settings is both challenging and impractical, as observed in other studies in Kenya’s informal settlements [11,33]. Responding to these challenges requires supporting communities in these settlements along with economic recovery and support programmes to buffer them from the unequal economic shocks they face [34]. Although some efforts have been made to support vulnerable families through governmental and philanthropic efforts, it is important to ensure there is a coordinated approach by government and private sector entities towards ensuring there is equitable universal access and coverage, and transparent systems in executing such programmes and initiatives [34]. It also requires recognizing and pooling in pre-existing initiatives by local communities and community-based organizations into the planning and response stages to harness their efforts in tackling informal settlement priority issues, such as food insecurity, water and sanitation needs, child health, gender, and social protection through a coordinated and integrated approach [11,35].

This study provides evidence of the magnitude of mental health problems experienced by the most vulnerable in Kenya’s urban informal settlements, which is pertinent information for mental healthcare, programmatic support, and policy responses. The high levels of depression and anxiety, exacerbated by COVID-19 related anxiety, in this highly vulnerable population raises concerns for mental healthcare and response, as well as development of the next generation. Findings from this study corroborate emerging evidence from other low- and middle-income countries (LMIC) on the state of mental health during the pandemic. A nationwide cross-sectional survey in Bangladesh estimated the prevalence of depressive symptoms was 33% [36], while in Soweto-South Africa the risk of depression among adults was 14.5% [37]. A recent cross-sectional survey conducted by our research team in Dagoretti’s informal settlement among caregivers of children below two years (n = 612) found levels of depression (PHQ) were 19% (manuscript under preparation). Other studies based in Kenya’s informal settlements have highlighted mental health problems are more likely to be experienced by women, and associated with factors, such as exposure to violence of various forms [34,38,39]. Research shows that caregivers distress and especially during a health crisis, impacts negatively on child developmental and behavioral outcomes [40,41,42]. Caregivers’ mental health challenges and exposure to violence during a pandemic, could limit their capacity to engage in interactive parenting practices to provide responsive care and a nurturing environment for their children to grow and thrive [40,43]. If left unchecked, children growing up in such exposed environments have a higher risk of developing various cognitive, behavioral, and emotional difficulties later in life [43]. Hence, an urgent call for the need for multi-pronged approaches to not only respond to caregiver mental health problems, but also address the multiple sources of risk, especially for the vulnerable sub-populations, such as women and young girls, who disproportionately face greater forms of risks that limits their growth and development.

With regard to perinatal mental health, our findings did not reveal any significant differences in mental health scores by pregnancy status compared with the general sample of caregivers. Although this was the case, it is important to highlight that poor mental health during pregnancy has numerous detrimental effects for both expectant women and the unborn babies [17,18], and it is likely that the pandemic exacerbates the risk for mental illness and sub-optimal perinatal health care [15]. For instance, there have been reports within the Kenyan context of decreased and delayed antenatal attendance arising from the fear of COVID-19 infection and restricted access to care due to lockdowns and imposed curfew hours that limit access to services within designated hours [44,45]. The implications of these are that some pregnant women who opt to delay seeking care may miss out on the opportunity of early screening for mental health symptoms during routine care [44].

Kenya’s mental healthcare remains underdeveloped in terms of numbers, distribution of skilled personnel and functional care and treatment facilities, and estimates of mental health burden are not regularly reported [36,46]. During the pandemic, there were efforts by the Government to put in place policies to address mental health crisis and response [47]; however, access and reach remains a challenge [46]. People with low socio-economic status, such as informal settlement residents, face challenges accessing mental healthcare. Within the study, we were able to link caregivers presenting with moderate and severe mental health problems to an independent counsellor, though we recognize this intervention may only be accessible to a limited population. With the clear evidence emerging of the desperate need for mental health support amongst urban informal settlement populations, our study recommends the following measures. Within informal settlements, there is need for mobile and outreach services to support extension and access to general and mental health services, under the mandate of the Ministry of Health. This is particularly important to address common barriers, such as long distances to access facility care, indirect and direct costs associated with seeking care [35]. In times of crisis with mitigation effects that negatively affect these populations, such as lockdowns and curfews, deploying mobile health teams with adequately staffed health professionals including mental health specialist to offer regularly scheduled community outreach health services can help towards reducing the health burden in such communities [46].

Secondly, use of telephonic, mobile, and digital mental health innovations are critical and necessary when physical access to facilities is limited in times of crisis. The increase in mobile telephone and internet coverage in the country and especially in urban informal settlements, provides an important platform to deliver low-cost mental health interventions to maximize access and coverage [4,46]. For instance, the roll-out of toll-free mental health hotlines and tele-counselling services, introduced by the Kenyan Ministry of Health during the pandemic [48], and reported in other studies [4,49], are promising measures aimed at increasing reporting, creating awareness of mental health issues and offering services remotely. Such interventions could also address barriers associated with seeking in-person mental healthcare for fear of stigmatization and promote uptake of services. However, such initiatives require to be closely monitored, operated by well trained and qualified personnel, while ensuring there is a functioning identification and referral system to formal healthcare.

Other suggestions put forward for mental health response during COVID-19 within the Kenyan context include: the need for preparation and allocation of adequate funding to a formal mental health response plan specific to the COVID-19 pandemic; training of community health workers/volunteers on provision of psychological first-aid support at household and community level, in order to enhance improved access to mental healthcare and support for those in need during the pandemic; and the need for assessments/surveys on the mental health impact in order to inform decision-making during the COVID-19 pandemic [44,46,50].

### 4.3. Methodological Strengths and Limitations

Our study provides an account of the magnitude of mental health problems during the COVID-19 pandemic among a relatively large sample of caregivers in three urban informal settlements of Kenya. The findings further provide baseline information on the state of mental health in these vulnerable populations during the pandemic and could be transferable to similar settings. The rapid cross-sectional design using telephonic approaches allowed for quick and timely data collection across different geographies during the pandemic. However, some limitations to this approach included difficulties accessing participants with network connectivity problems or those unreachable after several call attempts. Furthermore, due to the cross-sectional nature of the study, it was impossible to infer on causality in the associations which were observed. Given that we relied on self-reports on the data collection instruments administered, a potential source of bias could have been respondents’ providing socially desirable responses, which is a common methodological limitation for Likert-scale response options. The use of non-probability and purposive selection of participants through CHVs as proxies to support with participant identification based on the eligibility criteria communicated may have introduced some level of selection bias. Although this approach was inevitable due to the COVID-19 restrictions and directives to minimize in-person researcher-participant interactions, we worked with established networks in the community to ensure there was a good level of reach to households with children below five years. A challenge for random sampling approaches within informal settlements is the lack of a public repository of contact details for all residents. Lessons learnt for future remote data collection in similar circumstances could include prior generation of a sampling frame drawn from administrative and health information records, and use of existing research databases, while taking into account data sharing and protection regulations. Our study did not collect information on past or pre-existing mental health illnesses, rather we focused on mental health issues arising during the first year of the COVID-19 pandemic. The under-representation of male caregivers in the sample limits the generalizability of findings to this population, and in future studies a more targeted sampling strategy may be required to address the skewness in the sample by gender distribution.

## 5. Conclusions

This study highlights that caregivers living in urban informal settlements have experienced a high burden of mental health issues during the first year of the COVID-19 pandemic and that care for caregivers is critical to mitigate negative effects on their children. It is likely as Kenya and other countries experience repeated spikes of COVID-19 infections with attendant lockdown measures and continued disruption to people’s lives and livelihoods, the general and mental health of caregivers of young children will further deteriorate in absence of interventions. Therefore, there is an urgent need to accelerate access to effective low-cost interventions aimed at enhancing mental healthcare during and beyond the pandemic. Multi-sectoral approaches are needed to tackle the root causes of mental health problems in this population, including violence, discrimination, and economic disruptions.

## Figures and Tables

**Figure 1 ijerph-18-10092-f001:**
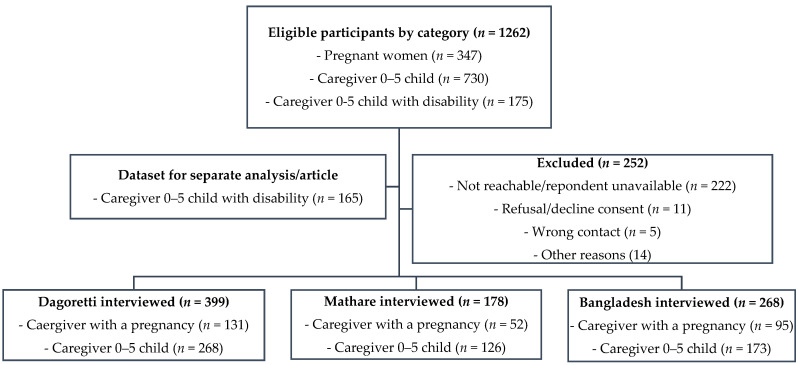
Participants’ recruitment summary.

**Table 1 ijerph-18-10092-t001:** A summary of socio-demographic variables stratified by participants’ mental health scores.

Characteristic	All Participants	PHQ-9			CAS (COVID-19 Anxiety Scale)			GAD-7 (General Anxiety Disorder)		
		No	Yes	*p*-Value	No	Yes	*p*-Value	No	Yes	*p*-Value
**Age (years) mean (SD)**	29.00 (7.15)									
<30	516 (61.07)	351 (63.02)	165 (57.29)	0.106	446 (61.35)	70 (59.32)	0.675	418 (61.65)	98 (58.68)	0.481
30 and above	329 (38.93)	206 (36.98)	123 (42.71)		281 (38.65)	48(40.68)		260 (38.35)	69 (41.32)	
**Gender**										
Female	801 (94.79)	531 (95.33)	270 (93.75)	0.326	686 (94.36)	115 (97.46)	0.160	645 (95.13)	156 (93.41)	0.370
Male	44 (5.21)	26 (4.67)	18 (6.25)		41 (5.64)	3 (2.54)		33 (4.87)	11 (6.59)	
**Marital status**										
Married or cohabiting	597 (71.17)	406 (73.15)	191 (67.25)	**0.012**	511 (70.68)	86 (74.14)	0.604	482 (71.30)	115 (70.55)	0.187
Single	119 (14.18)	82 (14.77)	37 (13.03)		106 (14.66)	13 (11.21)		101 (14.94)	18 (11.04)	
Separated/Divorced/Widowed	123 (14.66)	67 (12.07)	56 (16.72)		106 (14.66)	17 (14.66)		93 (13.76)	30 (18.40)	
**Education level**										
Primary school and below	376 (44.50)	233 (41.83)	143 (49.65)	0.086	315 (43.33)	61 (51.69)	0.098	301 (44.40)	75 (44.91)	0.975
Secondary School	379 (44.85)	260 (46.68)	119 (41.32)		329 (45.25)	50 (42.37)		304 (44.84)	75 (44.91)	
Tertiary	90 (10.65)	64 (11.49)	26 (9.03)		83 (11.42)	7 (5.93)		73 (10.77)	17 (10.18)	
**Occupation**										
Formal	42 (4.97)	30 (5.39)	12 (4.17)	0.32	40 (5.50)	2 (1.69)	0.08	30 (4.42)	12 (7.19)	0.32
Informal	388 (45.92)	246 (44.17)	142 (49.31)		325 (44.70)	63 (53.39)		311 (45.87)	77 (46.11)	
Unemployed	415 (49.11)	281 (50.45)	134 (46.53)		362 (49.79)	53 (443.92)		337 (49.710	78 (46.71)	
**Household asset index**										
Mean (SD)	(2.52) 1.19	2.65 (1.23)	2.26 (1.07)	**<0.001**	2.55 (1.22)	2.30 (1.02)	**0.030**	2.56 (1.21)	2.35 (1.09)	**0.047**
**Survey site**										
Dagoretti	396 (47.26)	284 (51.45)	112 (39.16)	**<0.001**	358 (49.72)	38 (32.20)	**<0.001**	327 (48.59)	69 (41.82)	**0.006**
Mathare	177 (21.12)	122 (22.10)	55 (19.23)		159 (22.08)	18 (15.25)		150 (22.29)	27 (16.36)	
Bangladesh	265 (31.62)	146 (26.45)	119 (41.61)		203 (28.19)	62 (52.54)		196 (29.12)	69 (41.82)	
**Pregnancy**										
Not pregnant	567 (67.10)	384 (68.94)	183 (63.54)	0.113	489 (67.26)	78 (66.10)	0.803	453 (66.81)	114 (68.26)	0.721
Pregnant	278 (32.90)	173 (31.06)	105 (36.46)		238 (32.74)	40 (33.90)		225 (33.19)	53 (31.74)	
Number of children ^+^										
0–3	623 (79.97)	406 (79.92)	217 (80.07)	0.960	536 (80.72)	87 (75.65)	0.210	506 (81.48)	117 (74.05)	**0.037**
>3	156 (20.03)	102 (20.08)	54 (19.93)		128 (19.28)	28 (24.35)		115 (18.52)	41 (25.95)	

Statistical significance (*p*-value ≤ 0.05) based on chi-square test of association for categorical variables and *t*-test for continuous variables. Abbreviations: HH-household; PHQ-9—patient health questionnaire; SD-standard deviation. Notes: ^+^ only 2 respondents had no children and the sample mean was 3.08. The cut-off is informed by the national average of 3.65 children per household.

**Table 2 ijerph-18-10092-t002:** COVID-19-related consequences on psychosocial aspects stratified by participants’ mental health scores.

Characteristics	All Participants	PHQ-9			CAS (COVID-19 Anxiety Scale)			GAD-7 (General Anxiety Disorder)		
		No	Yes	*p*-Value	No	Yes	*p*-Value	No	Yes	*p*-Value
**Experienced violence (COVID period)**										
Yes	107 (12.66)	47 (8.44)	60 (20.83)	**<0.001**	82 (11.28)	25 (21.19)	**0.003**	73 (10.77)	34 (20.36)	**0.001**
No	738 (87.34)	510 (91.56)	228 (79.17)		645 (88.72)	93 (78.81)		605 (89.23)	133 (79.64)	
**COVID-19 perceived as a serious threat**										
No	46 (5.46)	36 (6.47)	10 (3.48)	0.070	45 (6.20)	1 (0.58)	**0.018**	43 (6.35)	3 (1.81)	**0.021**
Yes	797 (94.54)	520 (93.53)	277 (96.52)		681 (93.80)	116 (99.15)		634 (93.65)	163 (98.19)	
**Experienced discrimination (COVID period)**										
Yes	129 (15.27)	57 (10.23)	72 (25.00)	**<0.001**	96 (13.20)	33 (27.97)	**<0.001**	79 (11.65)	50 (29.94)	**<0.001**
No	716 (84.73)	500 (89.77)	216 (75.00)		631 (86.80)	85 (72.03)		599 (88.35)	117 (70.06)	
**Experienced job loss (COVID period)**										
No or to a less extent	133 (17.69)	114 (23.41)	19 (7.17)	**<0.001**	123 (19.19)	10 (9.01)	**0.009**	115 (19.20)	18 (11.76)	**0.031**
Very much	619 (82.31)	373 (76.59)	246 (92.83)		518 (80.81)	101 (90.99)		484 (80.80)	135 (88.24)	
**Loss of income generation (COVID period)**										
No	129 (18.38)	102 (22.27)	27 (11.01)	**<0.001**	118 (19.80)	11 (10.38)	**0.021**	112 (20.04)	17 (11.89)	**0.025**
Yes	573 (81.62)	356 (77.73)	217 (88.93)		478 (80.20)	95 (89.62)		447 (79.96)	126 (88.11)	
**Ability to pay utilities affected (COVID period)**										
Not affected or affected to a less extent	135 (16.13)	92 (16.70)	43 (15.03)	0.535	111 (15.42)	24 (20.51)	0.165	107 (15.97)	28 (16.77)	0.802
Very much affected	702 (83.87)	459 (83.30)	243 (84.97)		609 (84.58)	93 (79.49)		563 (84.03)	139 (83.23)	
**Ability to repay loans affected (COVID period)**										
Not affected or affected to a less extent	158 (22.77)	99 (22.86)	59 (22.61)	0.937	131 (22.32)	27 (25.23)	0.508	124 (22.42)	34 (24.11)	0.669
Very much affected	536 (77.23)	334 (77.14)	202 (77.39)		456 (77.68)	80 (74.77)		429 (77.58)	107 (75.89)	
**Ability to meet basic childcare affected (COVID period)**										
Not affected or affected to a less extent	218 (27.39)	166 (31.98)	52 (18.77)	**<0.001**	193 (28.34)	25 (21.74)	0.142	188 (29.51)	30 (18.87)	**0.007**
Very much affected	578 (72.61)	353 (68.02)	225 (81.23)		488 (71.66)	90 (78.26)		449 (70.49)	129 (81.13)	
**Interaction outside households affected (COVID period)**										
Not affected or affected to a less extent	364 (43.23)	258 (46.40)	106 (37.06)	**0.010**	326 (44.97)	38 (32.48)	**0.011**	307 (45.41)	57 (34.34)	**0.010**
Very much affected	478 (56.77)	298 (53.60)	180 (62.94)		399 (55.03)	79 (67.52)		369 (54.59)	109 (65.66)	
**Family interactions affected (COVID period)**										
Not affected or affected to a less extent	290 (34.44)	212 (38.20)	78 (27.18)	**0.001**	259 (35.77)	31 (26.27)	**0.044**	248 (36.74)	42 (25.15)	**0.005**
Very much affected	552 (65.56)	343 (61.80)	209 (72.82)		465 (64.23)	87 (73.73)		427 (63.26)	125 (74.85)	

Statistical significance (*p*-value ≤ 0.05) based on chi-square test of association for categorical variables. Notes; PHQ-9—patient health questionnaire; SD—standard deviation.

**Table 3 ijerph-18-10092-t003:** Multivariate analysis of the relationship between socio-demographic factors, COVID-19-related consequences on psychosocial aspects, and participant’s mental health scores.

Characteristics	PHQ-9		CAS		GAD-7	
	Odds Ratio (95% CI)	*p*-Value	Odds Ratio (95% CI)	*p*-Value	Odds Ratio (95% CI)	*p*-Value
**Age**						
<30	1		1		1	
30 and above	1.15 (0.78–1.68)	0.487	0.80 (0.49–1.30)	0.360	0.77 (0.48–1.25)	0.291
**Sex**						
Female	1		1		1	
Male	1.18 (0.52–2.67)	0.686	0.44 (0.12–1.57)	0.205	1.37 (0.57–3.25)	0.482
**Marital status**						
Married or cohabiting	1		1		1	
Single	1.56 (0.88–2.76)	0.130	1.41 (0.70–2.86)	0.342	1.58 (0.82–3.04)	0.167
Separated/Divorced/Widowed	0.98 (0.57–1.71)	0.956	0.51 (0.24–1.08)	0.078	0.83 (0.44–1.55)	0.558
**Education Level**						
Tertiary			1		1	
Primary and below	1.39 (0.70–2.72)	0.345	2.79 (1.06–7.36)	**0.038**	0.99 (0.46–2.12)	0.984
Secondary School	1.22 (0.63–2.37)	0.563	2.24 (0.85–5.93)	0.103	1.02 (0.49–2.15)	0.952
**Occupation**						
Unemployed	1		1		1	
Formal	1.34 (0.58–3.12)	0.491	0.56 (0.12–2.58)	0.460	2.84 (1.21–6.63)	**0.016**
Informal	1.04 (0.70–1.54)	0.847	1.26 (0.78–2.05)	0.347	0.93 (0.59–1.49)	0.776
**Pregnancy**						
Not pregnant	1		1		1	
Pregnant	1.44 (0.96–2.16)	0.074	0.98 (0.59–1.63)	0.939	1.04 (0.63–1.69)	0.890
**No. of children**						
0–3	1		1		1	
>3	1.02 (0.61–1.68)	0.948	1.27 (0.72–2.25)	0.402	1.55 (0.93–2.57)	0.090
**Experienced violence** **(COVID period)**						
No	1		1		1	
Yes	1.89 (1.12–3.18))	0.016	1.83 (1.01–3.32)	**0.048**	1.80 (1.03–3.13)	**0.037**
**Perceived COVID-19 as a serious threat**						
No	1		1		1	
Yes	0.77 (0.31–1.88)	0.561	3.79 (0.49–29.43)	0.203	2.27 (0.51–10.10)	0.282
**Experienced discrimination** **(COVID period)**						
No	1		1		1	
Yes	3.12 (1.91–5.10)	**<0.001**	2.35 (1.35–4.09)	0.002	3.62 (2.19–5.98)	**<0.001**
**Experienced job loss** **(COVID period)**						
Not affected or affected to a less extent	1		1		1	
Very much affected	2.88 (1.51–5.48)	**0.001**	1.60 (0.69–3.70)	0.272	1.17 (0.58–2.39)	0.661
**Loss of income generation** **(COVID period)**						
No	1		1		1	
Yes	1.19 (0.68–2.15)	0.564	1.50 (0.67–3.38)	0.325	1.26 (0.63–2.53)	0.519
**Ability to pay utilities affected** **(COVID period)**						
Not affected or affected to a less extent	1		1		1	
Very much affected	0.45 (0.22–0.91)	**0.026**	0.31 (0.12–0.78)	**0.013**	0.34 (0.14–0.82)	**0.016**
**Ability to repay loans affected** **(COVID period)**						
Not affected or affected to a less extent	1		1		1	
Very much affected	0.64 (0.37–1.11)	0.112	0.87 (0.40–1.87)	0.721	0.48 (0.25–0.91)	**0.024**
**Ability to meet basic childcare affected** **(COVID period)**						
Not affected or affected to a less extent	1		1		1	
Very much affected	1.41 (0.88–2.24)	0.149	2.08 (0.87–4.98)	0.101	1.58 (0.90–2.75)	0.109
**Household asset index**	0.83 (0.69–0.99)	**0.041**	1.02 (0.81–1.28)	0.874	0.94 (0.76–1.16)	0.562
**Movement outside household affected** **(COVID period)**						
Not affected or affected to a less extent	1		1		1	
Very much affected	1.50 (0.98–2.31)	0.062	1.68 (0.96–2.93)	0.070	1.38 (0.84–2.29)	0.207
**Family interaction affected** **(COVID period)**						
Not affected or affected to a less extent	1		1		1	
Very much affected	1.16 (0.73–1.85)	0.518	1.11 (0.61–2.01)	0.739	1.17 (0.68–2.04)	0.565
**Survey study site**						
Dagoretti	1		1		1	
Mathare	1.02 (0.61–1.72)	0.934	1.11 (0.55–2.24)	0.772	0.56 (0.28–1.11)	0.098
Bangladesh	1.93 (1.26–2.98)	**0.003**	2.12 (1.23–3.65)	**0.007**	1.52 (0.93–2.48)	0.093

Statistical significance (*p*-value ≤ 0.05). Notes: CAS—COVID-19 anxiety scale; CI-confidence intervals; GAD-7—General Anxiety Disorder; PHQ-9—patient health questionnaire; SD—standard deviation.

## Data Availability

All data are available within the manuscript and Supplementary Files. Anonymized dataset related to this manuscript is available in an open data repository in Havard Dataverse [Angwenyi, Vibian, 2021, “Replication Data for: Mental Health during COVID-19 Pandemic among Caregivers of Young Children in Urban Informal Settlements of Nairobi and Mombasa. A Cross-sectional Telephone Survey”, https://doi.org/10.7910/DVN/WXNOPL, (uploaded on 2 June 2021) Harvard Dataverse]. Additional approvals will be required from the ethical review committees for applications to re-use data, which can be submitted to the Aga Khan Social Science and Humanities ERC (erc.ssha@aku.edu).

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
