# Peer review of "Mental Health during COVID-19 Pandemic among Caregivers of Young Children in Kenya’s Urban Informal Settlements. A Cross-Sectional Telephone Survey"

_ijerph, 2021, doi:10.3390/ijerph181910092_

Round 1
Reviewer 1 Report
I find the study really interesting but I think it needs more informationto be more precise.
It is not clear whether or not the survey is anonymous. It is important
to clarify this question because the article is determining whether or not a person has a
mental illness exist in caregivers.
Talking about the sample size with which it works, it is impossible to
know if it is representative of the population under study. On the other hand, I miss why the degree of depression is greater in
Bangladesh than in Dagoretti. Finally, it would be good to know if a similar study has been carried
out in those cities (Bangladesh, Mathare and Dagoretti) to know the level
of depression and anxiety of these caregivers before the pandemic.
Reviewer 2 Report
The manuscript titled "Mental Health during COVID-19 Pandemic among Caregivers of Young Children in Kenya's Urban Informal Settlements. A Cross-sectional Telephone Survey" addresses an important public health issue during the COVID-19 pandemic, namely the psychological impact of caregivers of children during a health emergence that imposed restrictive measures. Although the manuscript is well written and of interest to the scientific community, some adjustments are needed before it is worthy of publication in IJEPH:
- add to bibliography:
1) Fiorillo, A.; Gorwood, P. The consequences of the COVID-19 pandemic on mental health and implications for clinical practice. Eur Psychiatry 2020,63,32. doi: 10.1192/j.eurpsy.2020.35
2) Rubin, G.J.; Wessely, S. Coronavirus: The psychological effects of quarantining a city. BMJ 2020,368,m313. doi: https://doi.org/10.1136/bmj.m313
- Specify type of sampling (non probability, purposive?).
- Limitations: The authors report, correctly, several limitations of the study. In my opinion, the most serious limitation is the sampling procedure. The sample is purposive and the method of recruitment is not very clear as the eligibility criteria are clearly described.
Reviewer 3 Report
The authors reported on the impact of COVID-19 on anxiety & depression in caregivers in Kenya. This paper is very timely and important as the academic community continues to understand the consequences of COVID-19. However, I have made recommendations on methods and presentation of results that I believe will strengthen the paper. I appreciate the authors' efforts and hope they find my suggestions beneficial.
-I would encourage overall editing (e.g., typo on pg. 2)
INTRODUCTION
-The authors have provided adequate justification for the need to look into the impact of COVID-19 on mental health in vulnerable communities. The authors may consider providing more context for journal readers who may be less familiar with the Kenya's community (E.g., what is meant by "informal settlements?" on pg 2? What are "flying toilets" on pg 3?).
-The authors should provide more background on why it is important to study caregivers specifically. They briefly mention their concern on pg 2, but providing more context for why they chose to study this specific group is important (e.g., have others studies shown that caregivers are at increased risk for mental health concerns?, Are there specific vulnerabilities caregivers experience in these communities?, etc.)
-The authors referenced that this work draws on a larger study, but the citation provided is for a book on research design. It will be important for reviewers to understand how this paper is different from the larger study. This citation should be corrected and the authors should briefly discuss how this study is unique from the larger study and the data has not been unnecessarily divided.
MATERIALS/METHODS
-Please clarify if having children with disabilities was an exclusion criteria in this study, as it was not listed by noted to be reported in a different study (pg 3).
-The procedures in general aren't very clear. It would be helpful to discuss how many people were approached, how they were identified, what percentage declined/were contacted, how people completed the measures and over what time frame, etc.
-It would be helpful to describe each measure (what variable it was used to assess, the scale and any cutoff points, and the actual reliability and validity of the measures).
-I am concerned about the use of a scale (COVID-19 anxiety scale) that isn't validated. More importantly, I am unsure what is actually being evaluated by this scale--is it somatic symptoms from anxiety? I am unclear as to how scales measuring dizziness, sleep disturbance, appetite loss, etc. are assessing a person's anxiety and fear of the corona virus. I would recommend providing more background on this scale or removing it completely from the study.
-Please clarify the point values assigned to families using the SES index; was the score of one for assets, 3 for motor vehicles, etc. specific to this study or are those values assigned according to the 9-index instructions?
-Please provide more description of how the “survey instrument” was piloted and modified to its final version. Does “survey instrument” refer to all measures used in the study? How were the 14 respondents selected to pilot? How was the electronic questionnaire/survey instrument modified after the pilot?
-The authors have included a lot of variables in this study. It would be helpful to provide justification/background about why these specific variables were selected. In addition, I’m unclear if the variables are specific to COVID or are baseline concerns in the community. For example, they assessed if participants experienced violence, discrimination, ability to meet basic childcare, etc.—was this asking if the participants experienced these things in general, or just during the pandemic? Clarification is needed as this has implications on their results and conclusions. It may be beneficial to really focus on the impact specific to COVID, or at least better differentiate between COVID and pre-existing challenges/vulnerabilities.
-I am concerned about the number of chi square/t tests conducted as this can raise risk for Type 1 error. I encourage the authors to consult with a statistician to determine if multivariate analyses may be more appropriate for their aims.
-Please provide explanation for why those specific cut offs for the PHQ 9, COVID-19 anxiety scale, and generalized anxiety scales were selected.
RESULTS
-Please include the overall response rate to the survey
-It’s helpful to the reader when the actual p-value is reported, rather than noting it was less than .05 or .01
-It would be helpful to have a table that describes the measures’ scores by socio-demographics (e.g., X percent of females reported depression)
-It would be helpful to better explain the characteristics labels in tables (for example, what is meant by Job loss: no or less extent and very much?)
-I would also recommend the authors consult with a statistician to discuss presenting and interpreting results.
DISCUSSION
-The authors indicated that Kenyan’s experienced a “high burden” of mental health problems (pg 11), it would be helpful to provide context for this statement by including burden rates pre-pandemic (if available).
-Please revise the sentence beginning: “Findings show that a relatively higher household asset index….. on pg 11. As written, it sounds like the authors
-Some findings discussed in this section do not match the results reported (e.g., those unable to repay loans only had a lower likelihood of generalized anxiety per Table 3, but in this section it is reported lower likelihood for experiencing depression, gen anxiety, and COVID anxiety).
-The real strength of this study is the recommendations for mitigation of the increased mental health concerns they identified. This should be developed, and variables/results presented that specifically support this conclusion should be prioritized in this work. This would help increase clarity.
-The authors concluded that their findings show promise in engaging and working with community resource persons (pg 13), but they did not study this (it was part of their methods). This should be rephrased.
-In general, the authors did a good job describing the current state of these communities, but the variables contributing to the current state are less clear and would benefit from streamlining and additional description so that the reader can properly understand implications of their findings.
Round 2
Reviewer 3 Report
I appreciate the authors' extensive attention to reviewer feedback. They have sufficiently addressed my areas of concern and I feel the paper is much clearer.
Author Response
Dear Reviewer,
Thank you for your constructive feedback and are delighted to know all comments raised were addressed ssatisfactorily, and inproved the quality of the manuscript. We have rectified typos and grammatical errors that were in the previous manuscript submission.
Kind regards,
Vibian Angwenyi (on behalf of co-authors)